

# The relationship between serum uric acid within the normal range and β-cell function in Chinese patients with type 2 diabetes: differences by body mass index and gender

Xing Zhong*, Deyuan Zhang*, Lina Yang, Yijun Du and Tianrong Pan

Department of Endocrinology, The Second Affiliated Hospital of Anhui Medical University,
He Fei, Anhui Province, China
* These authors contributed equally to this work.

Corresponding author
Tianrong Pan,
pantianrong1968@163.com

## ABSTRACT

**Background:** Elevated serum uric acid (SUA) has a positive correlation with insulin secretion and insulin resistance indexes. However, whether weight- and gender-specific differences regarding the relationship between SUA within the normal range and β-cell function and insulin resistance exist is unknown in type 2 diabetes mellitus (T2DM) patients.
**Methods:** A total of 380 patients with type 2 diabetes were divided into two groups as overweight/obesity ($n$ = 268) and normal weight ($n$ = 112). Each group were again divided into low (LSUA) and high normal SUA (HSUA). The HbA1c, C-peptide, SUA, creatinine, and lipids profiles were measured. HOMA2IR and HOMA%2B were estimated using fasting glucose and C-peptide by homeostasis model assessment (HOMA). Pearson's correlations and multiple linear regression analyses were conducted to assess the associations between SUA levels and islet function indexes.
**Results:** In overweight/obesity subgroup, the levels of body mass index, fasting C-peptide (FCP), P2hCP, fasting CPI (FCPI), postprandial CPI (PPCPI), ΔC-peptide, HOMA2%B, and HOMA2IR were higher in HSUA group than in LSUA group. In contrast, the HbA1c, FBS, and P2hBS were lower in HSUA than in LSUA. In normal weight subgroup, there were no differences between the HSUA than LSUA group in terms of clinical characteristics. Pearson's correlations indicated that there were no significant correlations between SUA and insulin secretory capacity in normal weight group, but in overweight/obesity group, SUA had positive significant correlations with P2hCP, FCPI, PPCPI, ΔC-peptide, and HOMA2%B. In the female group, there were no significant correlations between SUA and insulin secretory capacity. However, in the male group, SUA had positive significant correlations with insulin secretory capacity include P2hCP, FCPI, PPCPI, ΔC-peptide, and HOMA2%B. Multiple linear regression showed that SUA was significantly associated with HOMA2%B, but not with HOMA2IR in overweight/obesity and male group.
**Conclusions:** Our study shows that SUA levels within normal range were associated with β-cell function in T2DM patients with overweight/obesity or male. This finding supports that the association between SUA within normal range and insulin secretion ability differs by weight and sex.

## INTRODUCTION

Type 2 diabetes mellitus (T2DM) has become a serious issue in China with increasing incidences over the past decades (*Ogurtsova et al., 2017*). Increasing evidence suggests that high serum uric acid (SUA) level is not only associated with metabolic syndrome (MS) (*Babio et al., 2015*), but also is regarded as a potential tool for early diagnosis of MS (*Chen et al., 2016*). Elevated the level of SUA is associated with increased risk of T2DM and prediabetes in individuals with normoglycaemia in a large population-based cohort study (*Dehghan et al., 2008*; *Van Der Schaft et al., 2017*). However, changes in SUA and blood glucose do not exhibit a linear relationship. SUA rises with increasing blood glucose concentrations in the normal and prediabetes population, while SUA levels are negatively associated with HbA1c in T2DM (*Kawamoto et al., 2018*).

Progressive deterioration of islet β-cell function and insulin resistance are considered as primary pathophysiological factors during the development of T2DM. SUA is the end product of an exogenous pool of pruines and endogenous purine metabolism, and the final oxidation product of purine metabolism in humans, which is responsible for the production of UA and damage of free radicals. In hyperuricemic subjects with IGT, the failure of β-cell function to compensate variation of insulin sensitivity, compared with non-hyperuricemic (*Simental-Mendia, Rodriguez-Moran & Guerrero-Romero, 2009*). Furthermore, elevated SUA harbors a positive correlation with insulin secretion and insulin resistance indexes in newly diagnosed T2DM patients (*Hu et al., 2018*), implying a possible role for SUA in β-cell function. However, the interaction of SUA within the normal range and body mass index (BMI) on β-cell function and insulin resistance in T2DM patients remains unknown.

Therefore, we investigated the relationship between SUA within the normal range and β-cell function as well as their potential confounding factors such as age, gender, diabetic duration, blood pressure, blood lipid profiles, renal function, and HbA1c by BMI and gender.

## MATERIALS AND METHODS

### Study subjects

A total of 380 patients with type 2 diabetes who visited the Second Affiliated Hospital of Anhui Medical University were randomly selected in this cross-sectional study. The diagnosis of T2DM was according to the criteria of the American Diabetes Association. The exclusion criteria were (1) with hyperuricemia defined as SUA ≥420 umol/L in men and ≥360 umol/L in women (*Fang & Alderman, 2000*), (2) with renal dysfunction defined as serum creatinine ≥106 umol/L in male and ≥97 umol/L in female or chronic kidney disease, (3) patients with severe pancreatic disease and liver disease and those who suffered recent diabetic ketoacidosis and hyperosmotic nonketotic diabetic coma. Written informed consent was provided by all participants. The study was approved by the Ethics Committee of the Second Affiliated Hospital of Anhui Medical University (approval number 2017027).

## Measurements

Study participants were asked about their age and family history. Body weight, height, and blood pressure were measured by the diabetic nurses. BMI was calculated by dividing weight (in kilograms) by the height (in meters) squared. Normal weight and overweight/obesity were defined as BMI <24 kg/m$^2$ and BMI ≥24 kg/m$^2$ for Chinese population, respectively, according to the Working Group on Obesity in China's BMI criterias (Hou et al., 2013; Zhou, 2002). Blood tests were carried out after an overnight fasting for glucose, serum total cholesterol (TC), triglycerides (TG), high-density lipoprotein cholesterol (HDL), low-density lipoprotein cholesterol (LDL), SUA, liver/renal functions and glycated hemoglobin (HbA1c).

After collecting fasting blood samples, subjects received a noodle mixed-meal in patients with T2DM. Blood samples were collected to measure the concentrations of glucose and C-peptide 2h after the meal. HOMA2IR and HOMA%2B were assessed using homeostasis model assessment based on paired of fasting plasma glucose (FPG) and FCP measurements (https://www.dtu.ox.ac.uk/homacalculator/) (Wallace, Levy & Matthews, 2004). Insulin secretory capacity was also evaluated by C-peptide index (CPI) and ΔC-peptide. Fasting CPI (FCPI) and postprandial CPI (PPCPI) were calculated by a ratio of serum C-peptide (nmol/L) to plasma glucose concentrations (mmol/L) at baseline and 2 h after meal. The value of ΔC-peptide was defined as increment in serum C-peptide level (nmol/L) at 2 h after the meal.

Serum C-peptide was measured by chemiluminescent enzyme immunoassay. HbA1c was measured by high-performance liquid chromatography. Plasma glucose was evaluated with the glucose oxidase method. TC, TG, HDL, LDL, SUA, and liver/renal functions were analyzed by the standardized enzymatic method.

## Statistical analyses

Continuous variables were expressed as means and standard deviation or medians and interquartiles. Categorical variables were expressed by numbers. In all the analyses, parameters with non-normal distributions were used after log transformation. For categorical variables, the Chi-square test was performed, while for continuous variables, Student $t$-test was used. Pearson's correlations were calculated to characterize the associations between islet function indexes and SUA levels within each group. To evaluate whether SUA was an independent risk factor for β-cell function in T2DM, we performed the multiple linear regression analysis. A two-tailed $p \leq 0.05$ was considered as statistically significant. All statistical analyses were conducted with SPSS software (Version 21.0).

## RESULTS

### Clinical and laboratory data of the patients according to BMI and SUA category

The characteristic of the study patients according to BMI was shown in Table 1. The levels of SBP, DBP, TG, FCP, P2hCP, FCPI, PPCPI, HOMA2%B, and HOMA2IR were higher in the overweight/obesity group than in normal weight group. Furthermore, the patients

**Table 1 Clinical characteristics and islet function indexes of T2DM patients by BMI.**

| Variables | Normal weight group (N = 112) | Overweight/obesity group (N = 268) | t/χ | p |
|---|---|---|---|---|
| SUA (umol/L) | 262.5 (224.3, 297.0) | 290.5 (256.0, 333.0) | −5.08 | <0.001 |
| Age (years) | 54.1 ± 11.9 | 52.1 ± 12.0 | 1.50 | 0.134 |
| Male/female | 63/49 | 171/97 | 1.38 | 0.168 |
| Duration (years) | 5.0 (1.0, 10.0) | 4.0 (0.3, 9.7) | 0.51 | 0.613 |
| SBP (mmHg) | 120.0 (110.0, 131.5) | 130.0 (120.0, 140.0) | −2.06 | 0.040 |
| DBP (mmHg) | 77.0 (70.0, 84.8) | 80.0 (76.0, 90.0) | 0.90 | <0.001 |
| BMI (kg/m$^2$) | 22.3 (20.6, 23.4) | 26.1 (25.4, 28.2) | −21.3 | <0.001 |
| TG (mmol/L) | 1.38 (0.88, 2.12) | 2.00 (1.22, 3.12) | −4.24 | <0.001 |
| TCH (mmol/L) | 4.37 (3.87, 5.11) | 4.54 (3.91, 5.20) | −1.01 | 0.315 |
| LDL (mmol/L) | 2.58 (2.18, 2.93) | 2.58 (2.18, 3.10) | 0.39 | 0.697 |
| HDL (mmol/L) | 1.07 (0.84, 1.38) | 1.01 (0.76, 1.10) | 2.86 | 0.004 |
| ALT (U/L) | 18.0 (14.0, 27.0) | 21.0 (15.0, 33.0) | −1.87 | 0.063 |
| CR (umol/L) | 68.5 (58.0, 81.8) | 73.0 (62.0, 85.0) | −1.73 | 0.084 |
| HbA1c (%) | 9.40 (7.53, 11.20) | 8.90 (7.60, 10.70) | 0.86 | 0.391 |
| FPG (mmol/L) | 9.49 ± 3.38 | 9.32 ± 3.03 | 0.47 | 0.637 |
| P2hPG (mmol/L) | 19.17 ± 4.91 | 18.69 ± 4.37 | 0.95 | 0.344 |
| FCP (nmol/L) | 1.84 (1.31, 2.82) | 2.40 (1.79, 3.31) | −4.28 | <0.001 |
| P2hCP (nmol/L) | 5.03 (3.52, 7.21) | 5.90 (4.13, 7.74) | −2.54 | 0.011 |
| FCPI | 0.22 (0.16, 0.32) | 0.28 (0.19, 0.37) | −3.77 | <0.001 |
| PPCPI | 1.49 (0.94, 2.35) | 1.78 (1.14, 2.62) | −2.24 | 0.026 |
| ΔC-peptide | 2.92 (1.76, 4.68) | 3.23 (1.90, 4.62) | −1.16 | 0.245 |
| HOMA2%B | 42.2 (28.0, 69.0) | 49.7 (33.9, 78.4) | −2.39 | 0.017 |
| HOMA2IR | 1.66 (1.17, 2.43) | 2.11 (1.60, 3.11) | 0.14 | <0.001 |

Note:
Values are expressed as mean ± standard deviation (SD) or median (range 25th–75th percentile).

were divided into two groups according the median SUA levels of patients with normal weight or overweight/obesity, respectively (low-normal SUA (LSUA) ≤285 umol/L; high-normal SUA (HSUA) >285 umol/L). In the overweight/obesity subgroup, the levels of BMI, ALT, CR, FCP, P2hCP, FCPI, PPCPI, ΔC-peptide, HOMA2%B, and HOMA2IR were higher in HSUA group than in LSUA group. In contrast, the HbA1c, FBS, P2hBS, and HDL were lower in HSUA than in LSUA (Table 2). In normal weight subgroup, there were on differences between the HSUA and LSUA group in terms of clinical characteristics (Table 2).

## Correlation between SUA and insulin secretory capacity within normal or overweight/obesity groups

The relationship between confounding factors including SUA and insulin secretory capacity within normal or overweight/obesity groups was shown in Table 3. In normal weight group, there were no significant correlations between SUA and insulin secretory capacity. However, in overweight/obesity group, FCP, P2hCP, FCPI, PPCPI, ΔC-peptide, HOMA2%B, and HOMA2IR correlated positively with SUA, while HbA1c correlated negatively with SUA. After adjusting for Cr, BMI, and gender, there were

**Table 2 Clinical characteristics and islet function indexes of overweight/obesity and normal weight group by the median of SUA.**

| Variables | Overweight/obesity group | | | | Normal weight group | | | |
|---|---|---|---|---|---|---|---|---|
| | LSUA | HSUA | t/χ | p | LSUA | HSUA | t/χ | p |
| SUA (umol/L) | <285 | 285–420 | | | <285 | 285–420 | | |
| Age (years) | 52.9 ± 11.2 | 51.4 ± 12.5 | 1.01 | 0.314 | 55.4 ± 10.9 | 51.7 ± 13.5 | 1.60 | 0.112 |
| Male/female | 62/56 | 109/41 | 11.58 | 0.001 | 38/34 | 25/15 | 0.99 | 0.320 |
| Duration (years) | 4.0 (0.3, 10.0) | 4.0 (0.29, 9.00) | 0.14 | 0.886 | 6.0 (1.0, 10.0) | 4.5 (0.42, 10.0) | −0.18 | 0.861 |
| SBP (mmHg) | 129.4 ± 16.3 | 128.7 ± 17.3 | 0.35 | 0.729 | 126.2 ± 17.1 | 122.8 ± 20.0 | 0.96 | 0.345 |
| DBP (mmHg) | 80.5 ± 10.2 | 82.1 ± 11.6 | −1.19 | 0.235 | 76.9 ± 9.5 | 77.1 ± 9.6 | −0.07 | 0.953 |
| BMI (kg/m$^2$) | 26.5 ± 1.9 | 27.4 ± 2.7 | −3.14 | 0.002 | 21.8 ± 1.9 | 21.6 ± 2.0 | −0.17 | 0.872 |
| TG (mmol/L) | 1.88 (1.09, 2.58) | 2.08 (1.34, 3.32) | −1.42 | 0.156 | 1.21 (0.84, 2.03) | 1.43 (1.00, 2.15) | −0.51 | 0.614 |
| TCH (mmol/L) | 4.46 (3.74, 5.35) | 4.57 (4.07, 5.15) | −0.83 | 0.407 | 4.37 (3.95, 5.08) | 4.33 (3.51, 5.26) | 0.53 | 0.595 |
| LDL (mmol/L) | 2.58 (2.19, 2.95) | 2.59 (2.17, 3.13) | −0.59 | 0.550 | 2.58 (2.31, 2.93) | 2.58 (2.02, 3.15) | −0.37 | 0.712 |
| HDL (mmol/L) | 1.07 ± 0.38 | 0.97 ± 0.40 | 2.35 | 0.020 | 1.24 ± 0.49 | 0.99 ± 0.29 | 2.94 | 0.004 |
| ALT (U/L) | 20.0 (14.0, 30.3) | 23.5 (17.0, 35.0) | −2.73 | 0.007 | 18.0 (14.3, 23.0) | 20.0 (14.0, 30.0) | −0.65 | 0.515 |
| CR (umol/L) | 70.9 ± 16.1 | 75.4 ± 14.9 | −2.53 | 0.012 | 70.2 ± 15.5 | 70.7 ± 14.8 | −0.16 | 0.872 |
| HbA1c (%) | 9.50 ± 2.13 | 8.89 ± 1.96 | 2.40 | 0.020 | 9.32 ± 2.32 | 9.71 ± 2.75 | −0.78 | 0.434 |
| FPG (mmol/L) | 9.7 ± 2.8 | 9.0 ± 3.2 | 2.16 | 0.032 | 9.5 ± 3.3 | 9.5 ± 3.5 | 0.08 | 0.931 |
| P2hPG (mmol/L) | 19.4 ± 3.9 | 18.1 ± 4.7 | 2.44 | 0.015 | 18.9 ± 4.9 | 19.5 ± 4.9 | −0.49 | 0.636 |
| FCP (nmol/L) | 2.24 (1.71, 3.02) | 2.50 (1.87, 3.41) | −2.52 | 0.012 | 1.81 (1.30, 2.74) | 1.92 (1.32, 3.09) | −0.87 | 0.388 |
| P2hCP (nmol/L) | 5.00 (3.63, 6.73) | 6.52 (4.87, 8.43) | −4.45 | <0.001 | 4.87 (3.20, 6.68) | 5.46 (3.58, 7.69) | −0.72 | 0.474 |
| FCPI | 0.24 (0.17, 0.34) | 0.31 (0.22, 0.42) | −3.82 | <0.001 | 0.22 (0.16, 0.30) | 0.25 (0.15, 0.36) | −0.88 | 0.381 |
| PPCPI | 1.46 (0.95, 2.36) | 2.04 (1.35, 2.95) | −4.52 | <0.001 | 1.45 (0.94, 2.18) | 1.76 (0.94, 2.60) | −0.36 | 0.716 |
| ΔC-peptide | 2.52 (1.44, 4.07) | 3.81 (2.28, 5.46) | −4.26 | <0.001 | 2.82 (1.60, 4.77) | 3.36 (1.77, 4.66) | −0.69 | 0.492 |
| HOMA2%B | 45.4 (30.3, 63.4) | 60.3 (37.6, 90.9) | −1.82 | <0.001 | 40.3 (29.2, 64.1) | 43.5 (26.7, 91.3) | −0.68 | 0.493 |
| HOMA2IR | 2.03 (1.53, 2.75) | 2.23 (1.62, 3.16) | −4.69 | 0.007 | 1.64 (1.17, 2.32) | 1.86 (1.12, 2.66) | −0.71 | 0.477 |

**Note:**
Values are expressed as mean ± standard deviation (SD) or median (range 25th–75th percentile).

no significant correlations between SUA and HOMA2IR. After additional adjustment for HbA1c and Duration, SUA still had positive significant correlations with insulin secretory capacity include P2hCP, FCPI, PPCPI, ΔC-peptide, and HOMA2%B.

To further define the relation between SUA and HOMA2%B in overweight/obesity group, multiple linear regression was carried out using SUA as the dependent variable (Table 4). FCP, P2hCP, FCPI, PPCPI, and ΔC-peptide were excluded from the model because of their high correlation with HOMA2%B. FBS and P2hBS were also excluded because of their high correlation with HbA1c. SUA levels were significantly associated with HOMA2%B in unadjusted analyses. After adjustments for sex, Cr, BMI, HbA1c, and Duration, SUA remained positively associated with HOMA2%B.

## Clinical and laboratory data of the patients according to gender and SUA category

The characteristic of the study patients according to gender was shown in Table 5. There were 234 males and 146 females. The male group was younger and had a shorter
**Table 3 Correlation of selected variables with SUA in T2DM patients with overweight/obesity group.**

|  | Crude | | Adjusted for Cr, BMI, sex | | Adjusted for Cr, BMI, sex, HbA1c, Duration | |
| --- | --- | --- | --- | --- | --- | --- |
|  | r | p | r | p | r | p |
| HbA1c | −0.186 | 0.002 | −0.226 | <0.001 | | |
| FCP | 0.194 | 0.001 | 0.130 | 0.034 | 0.115 | 0.085 |
| P2hCP | 0.286 | <0.001 | 0.274 | <0.001 | 0.220 | 0.001 |
| FCPI | 0.268 | <0.001 | 0.222 | <0.001 | 0.142 | 0.034 |
| PPCPI | 0.308 | <0.001 | 0.296 | <0.001 | 0.232 | <0.001 |
| ΔC-peptide | 0.255 | <0.001 | 0.275 | <0.001 | 0.215 | 0.001 |
| HOMA2%B | 0.257 | <0.001 | 0.235 | <0.001 | 0.137 | 0.040 |
| HOMA2IR | 0.142 | 0.020 | 0.082 | 0.158 | 0.105 | 0.117 |

**Table 4 Multiple linear regression analysis for SUA and HOMA2%B in T2DM patients with overweight/obesity.**

|  | Partial regression coefficient (B) | Standard error (SE) | Standard partial regression coefficient (β) | t | p-Value |
| --- | --- | --- | --- | --- | --- |
| HOMA2%B (unadjusted) | 0.076 | 0.018 | 0.257 | 4.337 | <0.001 |
| HOMA2%B (adjusted for model 1: sex, Cr, BMI) | 0.066 | 0.017 | 0.223 | 3.930 | <0.001 |
| HOMA2%B (adjusted for model 2: model 1, HbA1c and Duration) | 0.049 | 0.022 | 0.182 | 2.135 | 0.013 |

diabetic duration compared to the female group. Compared with the female group, the levels of SUA, ALT, and CR in the male group were higher. Furthermore, the patients were divided into two groups according to the median SUA levels of patients in the male (LSUA ≤292.0 umol/L; HSUA >292.0 umol/L) or female (LSUA ≤264.5 umol/L; HSUA >264.5 umol/L) group, respectively (Table 6). In the male subgroup, the levels of BMI, ALT, HbA1c, P2hCP, FCPI, PPCPI, ΔC-peptide, and HOMA2%B were higher in HSUA group than in LSUA group. In contrast, the HbA1c, FBS, and P2hBS were lower in HSUA than in LSUA. In the female subgroup, the levels of BMI, TG, CR, and HOMA2IR were higher in HSUA group than in LSUA group.

## Correlation between SUA and insulin secretory capacity by gender category

The relationship between confounding factors including SUA and insulin secretory capacity within male or female groups was shown in Table 7. In male group, FCP, P2hCP, FCPI, PPCPI, ΔC-peptide, and HOMA2%B correlated positively with SUA, while HbA1c correlated negatively with SUA. After adjusting for Cr and BMI, there were also significant correlations between SUA and HOMA2IR. After additional adjustment for HbA1c and Duration, SUA still had positive significant correlations with insulin secretory

**Table 5 Clinical characteristics and islet function indexes of T2DM patients by gender.**

| Variables | Male (N = 234) | Female (N = 146) | t/Z | p |
|---|---|---|---|---|
| SUA (umol/L) | 292.0 (256.0, 339.5) | 264.5 (233.5, 297.0) | 5.01 | <0.001 |
| Age (years) | 49.9 ± 12.3 | 57.2 ± 9.9 | −6.03 | <0.001 |
| Duration (years) | 3.3 (0, 8.0) | 5.5 (1.0, 10.0) | −3.38 | 0.001 |
| SBP (mmHg) | 128.0 (114.8, 136.5) | 130.0 (118.0, 140.0) | −0.91 | 0.363 |
| DBP (mmHg) | 80.0 (75.5, 90.0) | 80.0 (70.0, 84.5) | −3.24 | 0.001 |
| BMI (kg/m$^2$) | 25.5 (23.8, 27.7) | 25.4 (23.4, 27.3) | −1.71 | 0.088 |
| TG (mmol/L) | 1.99 (1.12, 3.13) | 1.54 (0.96, 2.29) | −2.89 | 0.004 |
| TCH (mmol/L) | 4.44 (3.87, 5.20) | 4.52 (3.94, 5.15) | −0.32 | 0.752 |
| LDL (mmol/L) | 2.58 (2.09, 3.10) | 2.58 (2.34, 3.01) | −1.23 | 0.220 |
| HDL (mmol/L) | 0.95 (0.74, 1.07) | 1.07 (0.92, 1.34) | −4.61 | <0.001 |
| ALT (U/L) | 21.5 (16.0, 35.3) | 18.0 (13.0, 26.9) | −3.68 | <0.001 |
| CR (umol/L) | 75.5 (64.0, 87.3) | 67.0 (56.8, 76.6) | −4.89 | <0.001 |
| HbA1c (%) | 9.22 (7.98, 10.70) | 8.45 (7.00, 11.2) | −1.97 | 0.051 |
| FBS (mmol/L) | 9.50 ± 2.99 | 9.16 ± 3.35 | 1.04 | 0.301 |
| P2hBS (mmol/L) | 18.85 ± 4.36 | 18.80 ± 4.81 | 0.11 | 0.913 |
| FCP (nmol/L) | 2.25 (1.69, 3.27) | 2.26 (1.58, 3.03) | −1.33 | 0.182 |
| P2hCP (nmol/L) | 5.53 (3.94, 7.34) | 5.74 (4.21, 8.04) | −0.94 | 0.346 |
| FCPI | 0.27 (0.18, 0.36) | 0.26 (0.17, 0.35) | −0.43 | 0.671 |
| PPCPI | 1.61 (1.04, 2.45) | 1.71 (1.01, 2.85) | −0.91 | 0.365 |
| ΔC-peptide | 2.97 (1.74, 4.30) | 3.29 (1.95, 5.40) | −1.97 | 0.053 |
| HOMA2%B | 47.3 (31.6, 75.1) | 52.3 (30.5, 79.9) | −0.75 | 0.471 |
| HOMA2IR | 2.04 (1.47, 3.09) | 2.02 (1.31, 2.63) | 0.45 | 0.140 |

Note:
Values are expressed as mean ± standard deviation (SD) or median (range 25th–75th percentile).

capacity include P2hCP, FCPI, PPCPI, ΔC-peptide, and HOMA2%B. However, in female group, SUA only correlated positively with P2hCP and ΔC-peptide.

To further define the relation between SUA and HOMA2%B or HOMA2IR, multiple linear regression was carried out using SUA as the dependent variable (Table 8). In male group, SUA levels were significantly associated with HOMA2%B in unadjusted analyses. After adjustments for Cr, BMI, HbA1c and Duration, SUA remained positively associated with HOMA2%B. SUA levels were significantly associated with HOMA2IR in unadjusted analyses. After adjustments for Cr, BMI, HbA1c and Duration, there were no significant correlations between SUA and HOMA2IR. In contrast, there were no significant correlations between SUA and HOMA2%B and HOMA2IR in the female group.

## Correlation between islet function/insulin resistance and related variables in T2DM patients

To identify confounding factors affecting islet function and insulin resistance, a multiple linear regression was again performed in T2DM patients. Independent variables such as SUA, age, gender, diabetic duration, SBP, DBP, BMI, TG, TCH, LDL, HDL, ALT, CR, HbA1c were enrolled (Table 9). HOMA2%B had positive associations with BMI, SUA, age

**Table 6 Clinical characteristics and islet function indexes of male and female group by the median of SUA.**

| Variables | Male group ($n = 234$) | | | | Female group ($n = 146$) | | | |
|---|---|---|---|---|---|---|---|---|
| | LSUA <292 umol/L | HSUA ≥292 umol/L | t/Z | p | LSUA <264.5 umol/L | HSUA ≥264.5umol/L | t/Z | p |
| Age (years) | 50.78 ± 12.79 | 48.91 ± 11.76 | 1.16 | 0.247 | 55.90 ± 9.27 | 58.3 ± 10.49 | −1.44 | 0.153 |
| Duration (years) | 4.0 (0.1, 8.0) | 3.0 (0.0, 7.0) | −0.76 | 0.447 | 5.0 (1.0, 10.0) | 6.0 (1.3, 10.0) | −0.70 | 0.481 |
| SBP (mmHg) | 128.0 (115.0, 136.0) | 126.0 (114.0, 138.0) | −0.35 | 0.726 | 128.0 (118.0, 136.0) | 130.0 (120.0, 140.0) | −1.26 | 0.209 |
| DBP (mmHg) | 80.0 (74.0, 90.0) | 80.0 (76.0, 90.0) | −0.45 | 0.685 | 76.0 (70.0, 80.0) | 80.0 (70.0, 88.0) | −1.20 | 0.229 |
| BMI (kg/m$^2$) | 25.4 (23.2, 26.6) | 25.9 (24.5, 28.4) | −3.44 | 0.001 | 24.9 (22.5, 26.1) | 25.5 (23.6, 27.9) | −2.02 | 0.044 |
| TG (mmol/L) | 1.93 (1.02, 2.59) | 2.01 (1.27, 3.35) | −1.69 | 0.089 | 1.43 (0.91, 1.91) | 1.82 (1.03, 2.72) | −2.21 | 0.027 |
| TCH (mmol/L) | 4.38 (3.80, 5.19) | 4.57 (3.92, 5.18) | −0.99 | 0.319 | 4.55 (3.95, 5.22) | 4.43 (3.91, 5.05) | −0.55 | 0.585 |
| LDL (mmol/L) | 2.58 (2.07, 3.10) | 2.58 (2.15, 3.12) | −0.88 | 0.380 | 2.58 (2.33, 2.99) | 2.58 (2.34, 3.02) | −0.36 | 0.720 |
| HDL (mmol/L) | 0.98 (0.74, 1.10) | 0.91 (0.76, 1.07) | −0.85 | 0.395 | 1.10 (1.01, 1.63) | 1.07 (0.81, 1.23) | −3.38 | 0.001 |
| ALT (U/L) | 18.0 (14.0, 31.0) | 25.0 (18.0, 42.0) | −3.48 | <0.001 | 19.0 (13.0, 26.5) | 18.0 (13.5, 27.0) | −0.42 | 0.676 |
| CR (umol/L) | 76.0 (63.0, 88.0) | 74.0 (65.0, 86.0) | −0.03 | 0.978 | 62.0 (52.5, 74.0) | 72.0 (61.5, 78.5) | −2.95 | 0.003 |
| HbA1c (%) | 9.90 (8.30, 11.30) | 8.80 (7.70, 10.0) | 4.30 | <0.001 | 8.10 (6.85, 10.65) | 8.73 (7.02, 11.42) | −1.18 | 0.237 |
| FBS (mmol/L) | 10.09 ± 2.82 | 8.89 ± 3.05 | 3.10 | 0.002 | 8.75 ± 3.17 | 9.57 ± 3.49 | −1.50 | 0.137 |
| P2hBS (mmol/L) | 19.78 ± 3.99 | 17.88 ± 4.55 | 3.42 | 0.001 | 18.29 ± 4.76 | 19.32 ± 4.82 | −1.33 | 0.186 |
| FCP (nmol/L) | 2.20 (1.65, 2.91) | 2.46 (1.73, 3.60) | −1.95 | 0.051 | 2.11 (1.29, 2.85) | 2.40 (1.78, 3.23) | −1.99 | 0.046 |
| P2hCP (nmol/L) | 4.85 (3.54, 6.67) | 6.19 (4.48, 8.06) | −3.95 | <0.001 | 5.41 (3.79, 7.76) | 6.17 (4.45, 8.65) | −1.48 | 0.138 |
| FCPI | 0.22 (0.17, 0.33) | 0.31 (0.22, 0.47) | −3.93 | <0.001 | 0.25 (0.16, 0.33) | 0.26 (0.17, 0.41) | −0.86 | 0.392 |
| PPCPI | 1.35 (0.92, 1.96) | 2.03 (1.41, 2.69) | −4.53 | <0.001 | 1.69 (1.05, 2.76) | 1.84 (0.99, 3.08) | −0.91 | 0.362 |
| ΔC-peptide | 2.32 (1.38, 3.84) | 3.55 (2.26, 5.27) | −4.01 | <0.001 | 3.08 (1.84, 5.25) | 3.75 (2.19, 5.51) | −1.06 | 0.288 |
| HOMA2%B | 38.3 (27.9, 59.8) | 59.3 (37.3, 89.0) | −4.39 | <0.001 | 56.5 (37.2, 74.6) | 48.7 (27.2, 86.7) | −0.27 | 0.784 |
| HOMA2IR | 1.98 (1.49, 2.65) | 2.20 (1.43, 3.25) | −1.24 | 0.214 | 1.81 (1.19, 2.49) | 2.18 (1.54, 2.97) | −2.38 | 0.017 |

Note:
Values are expressed as mean ±standard deviation (SD) or median (range 25th–75th percentile).

and diabetic duration and a negative correlation with HbA1c. HOMA2IR had positive associations with BMI and TG and a negative correlation with diabetic duration.

## DISCUSSION

In this study, we confirmed that SUA levels are significantly associated with HOMA2%B in T2DM patients with overweight/obesity and male group, but not in normal weight and female group. In addition, we also demonstrated that other islet function indexes, such as FCPI, PPCPI, and ΔC-peptide, did correlate with SUA levels in T2DM patients with overweight/obesity and male group. However, our study observed the absence of a relationship between SUA and HOMA2IR after adjustment for Cr, BMI, sex, HbA1c, and diabetic duration in T2DM patients with overweight/obesity or male. To the best of our knowledge, this study is the first that these effects of SUA within the normal range on determinants of β-cell function and insulin resistance in T2DM by BMI and gender categories.

Uric acid is the end product of purine metabolism and derives from the conversion of hypoxanthine to xanthine and of xanthine to uric acid. We observed that SUA was higher in T2DM patients with overweight/obesity group than in those with normal weight

**Table 7 Correlation of SUA with selected variables in T2DM patients with male and female group.**

| | | Crude | | Adjusted for Cr, BMI | | Adjusted for Cr, BMI, HbA1c, Duration | |
|---|---|---|---|---|---|---|---|
| | | r | p | r | p | r | p |
| Male group | HbA1c | −0.291 | <0.001 | −0.284 | <0.001 | | |
| | FCP | 0.235 | 0.001 | 0.142 | 0.031 | 0.101 | 0.127 |
| | P2hCP | 0.331 | <0.001 | 0.280 | <0.001 | 0.163 | 0.013 |
| | FCPI | 0.356 | <0.001 | 0.288 | <0.001 | 0.178 | 0.007 |
| | PPCPI | 0.351 | <0.001 | 0.322 | <0.001 | 0.195 | 0.003 |
| | ΔC-peptide | 0.293 | <0.001 | 0.273 | <0.001 | 0.147 | 0.026 |
| | HOMA2%B | 0.350 | <0.001 | 0.319 | <0.001 | 0.195 | 0.003 |
| | HOMA2IR | 0.156 | 0.017 | 0.065 | 0.322 | 0.066 | 0.320 |
| Female group | HbA1c | 0.013 | 0.876 | 0.011 | 0.884 | | |
| | FCP | 0.165 | 0.046 | 0.108 | 0.199 | 0.113 | 0.182 |
| | P2hCP | 0.203 | 0.014 | 0.182 | 0.029 | 0.200 | 0.017 |
| | FCPI | 0.171 | 0.039 | 0.137 | 0.101 | 0.155 | 0.066 |
| | PPCPI | 0.135 | 0.104 | 0.114 | 0.175 | 0.133 | 0.116 |
| | ΔC-peptide | 0.182 | 0.028 | 0.177 | 0.034 | 0.198 | 0.018 |
| | HOMA2%B | 0.134 | 0.106 | 0.135 | 0.108 | 0.163 | 0.053 |
| | HOMA2IR | 0.149 | 0.072 | 0.090 | 0.282 | 0.094 | 0.268 |

**Table 8 Multiple linear regression analysis for SUA and HOMA2%B or HOMA2IR in T2DM patients by gender category.**

| | | Partial regression coefficient (B) | Standard error (SE) | Standard partial regression coefficient (β) | t | p-Value |
|---|---|---|---|---|---|---|
| Male group | HOMA2%B | | | | | |
| | Unadjusted | 0.514 | 0.090 | 0.350 | 5.69 | <0.001 |
| | Adjusted for model 1: Cr, BMI | 0.458 | 0.090 | 0.312 | 5.10 | <0.001 |
| | Adjusted for model 2: model 1, HbA1c and Duration | 0.319 | 0.107 | 0.217 | 2.99 | 0.003 |
| | HOMA2IR | | | | | |
| | Unadjusted | 2.986 | 3.323 | 0.156 | 2.40 | 0.017 |
| | Adjusted for model 1: Cr, BMI | 3.415 | 3.443 | 0.067 | 0.99 | 0.322 |
| | Adjusted for model 2: model 1, HbA1c and Duration | 3.346 | 3.359 | 0.065 | 0.99 | 0.320 |
| Female group | HOMA2%B | | | | | |
| | Unadjusted | 0.141 | 0.087 | 0.134 | 1.626 | 0.106 |
| | Adjusted for model 1: Cr, BMI | 0.137 | 0.085 | 0.131 | 1.618 | 0.108 |
| | Adjusted for model 2: model 1, HbA1c and Duration | 0.197 | 0.101 | 0.188 | 1.949 | 0.053 |
| | HOMA2IR | | | | | |
| | Unadjusted | 4.703 | 2.593 | 0.149 | 1.814 | 0.072 |
| | Adjusted for model 1: Cr, BMI | 2.783 | 2.578 | 0.088 | 1.079 | 0.282 |
| | Adjusted for model 2: model 1, HbA1c and Duration | 2.940 | 2.646 | 0.093 | 1.111 | 0.268 |

**Table 9 Multiple linear regression analysis on related variables for islet function indexes in T2DM patients.**

| | | Partial regression coefficient (B) | Standard error (SE) | Standard partial regression coefficient (β) | p-Value |
|---|---|---|---|---|---|
| HOMA2%B | HbA1c | −9.103 | 0.781 | −0.501 | <0.001 |
| | SUA | 0.127 | 0.032 | 0.177 | <0.001 |
| | Age | 0.486 | 0.159 | 0.146 | 0.002 |
| | BMI | 1.143 | 0.522 | 0.095 | 0.029 |
| | Duration | −0.697 | 0.327 | −0.100 | 0.034 |
| HOMA2IR | BMI | 0.089 | 0.018 | 0.241 | <0.001 |
| | TG | 0.076 | 0.023 | 0.165 | 0.001 |
| | Duration | −0.029 | 0.010 | −0.134 | 0.006 |

group, SUA within normal range independently related to obesity in T2DM. Consistent with our results, several previous studies have also shown the relationship between BMI and uric acid (*Han et al., 2018*). For example, *Chen et al. (2017)* also found that prevalence of obesity steadily increased across SUA quartiles in T2DM. A 10-year follow-up study demonstrated that BMI had a significant independent association with uric acid in all race-sex-groups (*Rathmann et al., 2007*). Furthermore, in subjects without diabetes or hyperuricemia, SUA levels were also associated with BMI, waist circumference, and waist-to-hip ratio (*Jin et al., 2013*). Interestingly, *Zhou et al. (2017)* found that successful weight control, mostly >10 kg weight reduction, was correlated with significant uric acid reduction after 2 years observation. Therefore, SUA levels, even in normal range, were associated with BMI in T2DM patient.

In addition to strong association with BMI, SUA is also associated with β-cell function in T2DM. *Tang et al. (2014)* found that patients with higher levels of SUA had higher insulin secretion, including the early phase and total insulin secretion in T2DM patients. Similarly, another study (*Hu et al., 2018*) has also reported that SUA augments insulin secretion, particularly basal insulin secretion, in the population-based study of newly diagnosed T2DM. Even in nondiabetic population, higher SUA levels also significantly correlate with lower early-phase insulin secretion (*Shimodaira et al., 2014*). However, the abovementioned studies do not evaluate the relationship between SUA in the normal range and β-cell function. Most of prior studies researching the association between SUA and β-cell function did not conduct subgroup analyses by BMI categories. Our present results show that SUA in the normal range is significantly associated with HOMA2%B in T2DM patients with overweight/obesity, but not in the normal weight group. Although it is not possible to explain the mechanism underlying this body weight difference from our study, this observation may be due to the influence of SUA levels, which our study showed that SUA levels were higher in T2DM patients with overweight/obesity than in those with normal weight group. Although subjects with higher SUA secrete more insulin, it does not mean that high SUA is beneficial to β-cell function. SUA becomes a strong oxidant in the environment of obesity (*Johnson et al., 2009*), which may in turn

promote lipid oxidation. In addition, obesity is related to elevated SUA level via both low urinary urate excretion and overproduction of SUA (*Matsuura et al., 1998*). A recent study found that an elevated level of uric acid causes β-cell injury via the NFκB-iNOS-NO signaling axis (*Jia et al., 2013*). Furthermore, *Sun et al. (2015)* found that uric acid-associated genes have an impact on insulin secretion in a Chinese patients with T2DM. Finally, another study (*Seyed-Sadjadi et al., 2017*) showed that the associations between SUA and diabetes risk factors are largely dependent on visceral fat mass in a non-diabetic population. Physicochemical properties define hyperuricemia as levels above the solubility threshold (6.8 mg/dL). With regard to metabolic sequel, high-normal SUA levels are already associated with an increased risk in patient with overweight/obesity.

The disposition index (DI) is thought to reflect the capacity for insulin secretion adjusted for insulin sensitivity and thus to provide a useful measure of β-cell function. PP-CPI, a ratio of the circulating level of C-peptide to that of glucose, is correlated with clamp DI (*Okuno et al., 2013*). In the present study, we found that PPCPI and ΔC-peptide had positive associations with SUA levels in overweight/obesity group, but not in normal weight group. Our findings agree with previous report by *Tang et al. (2014)*, which shows that patients with higher SUA had greater disposition indices (both DI30 and DI120). Taken together, accumulated evidence suggest SUA levels may be associated with insulin secretion in T2DM patients with overweight/obesity.

Another important finding in our study was that SUA had positive significant correlations with insulin secretory capacity include P2hCP, FCPI, PPCPI, ΔC-peptide, and HOMA2%B in male group. Hyperuricemia affected men more commonly than women. There was a SUA difference of 30–120 umol/L between men and women (*Akizuki, 1982*). It is previously known that estrogen may promote excretion of uric acid (*Hu et al., 2018*). Together, these result indicate that gender differences in association between SUA within normal range and insulin secretion in patients with T2DM. However, a previous study (*Hu et al., 2018*) suggested that elevated SUA was associated with insulin secretion in male and female. The mechanism underlying this sex-based difference remains unclear, and requires further study.

The evidence of the linkage between SUA and insulin resistance in type 2 diabetes is growing, but it is unclear if SUA within the normal range directly lead to declines in insulin sensitivity in T2DM patients. However, our study observed the absence of a relationship between SUA within normal range and insulin resistance in T2DM patients with overweight/obesity and normal weight groups. Other researchers (*Hu et al., 2018*; *Wang et al., 2011*) have also demonstrated that the UA levels of hyperuricemic patients have no effect on their insulin sensitivity index. *Liu & Ho (2011)* study suggested that SUA was not associated with insulin resistance after adjustment for BMI, TG, and BP. There are several possible explanations for the lack of independent relationship between SUA within normal range and insulin resistance in this study. Firstly, this result could be driven by SUA levels that are well within the normal range. Secondly, these discrepancies could be related the techniques used for measurement of insulin sensitivity. Finally, UA has an important role as an antioxidant (*Lippi et al., 2008*), but elevated SUA may cause oxidative stress (*Pasalic, Marinkovic & Feher-Turkovic, 2012*) and inhibit

endothelial NO bioavailability (*Sharaf El Din, Salem & Abdulazim, 2017*), all of which closely associated with the insulin resistance. Collectively, the exact role of SUA within normal range in oxidation is still worth further investigation in T2DM patients.

The relationship between SUA and HbA1c has been reported. For example, *Kawamoto et al. (2018)* found a negative association between SUA and HbA1c was shown particularly in men with HbA1c ≥6.5%. *Cui et al. (2016)* showed that a negative correlation between uric acid and HbA1c is conditional in newly diagnosed type 2 diabetes patients. In our study, we also found that SUA within normal range negatively related to HbA1c in T2DM patients with overweight/obesity. In T2DM patients with normal weight group, the partial correlation analysis demonstrated the negative correlation between SUA and HbA1c, but no significant difference was observed with multiple linear regression analysis. These results indicated that there was negatively association between SUA, even within normal range, and HbA1c in T2DM patients with overweight/obesity.

Unfortunately, this study has some limitations. Firstly, we do not analyze whether oral hypoglycemic agents have an effect on SUA. Sodium-glucose co-transporter 2 inhibitor (SGLT-2i) could improve glycemic control and lower SUA levels in T2DM (*Hao et al., 2018*). However, other hypoglycemic drugs, including metformin, rosiglitazone, glibenclamide, and pharmacologic insulin, do not have a large impact on SUA concentration (*Hussain et al., 2018*; *Iliadis et al., 2007*; *MacFarlane, Liu & Solomon, 2015*). In our study, the T2DM patients were treated with oral hypoglycemic drugs (not including SGLT-2i) and insulin. Secondly, the number of subjects enrolled was relatively small. Thirdly, the relationship between SUA within normal range and oxidative stress is still worth further investigation in T2DM.

## CONCLUSION

Our study shows that SUA levels within normal range are associated with β-cell function in T2DM patients with overweight/obesity, and the relationship also displays sex-based differences. However, SUA levels within normal range are not related to insulin resistance in T2DM patients. This finding supports the association between SUA within normal range and insulin secretion ability differs by weight and gender.

### Funding
The authors received no funding for this work.

### Competing Interests
The authors declare that they have no competing interests.

### Author Contributions
- Xing Zhong performed the experiments, analyzed the data.
- Deyuan Zhang performed the experiments, prepared figures and/or tables, authored or reviewed drafts of the paper.
- Lina Yang performed the experiments, contributed reagents/materials/analysis tools.

- Yijun Du analyzed the data.
- Tianrong Pan conceived and designed the experiments, approved the final draft.

## Human Ethics

The following information was supplied relating to ethical approvals (i.e., approving body and any reference numbers):

The Ethics Committee of the Second Affiliated Hospital of Anhui Medical University granted Ethical approval to carry out the study within its facilities (Ethical Application Ref: 2017027).

## Data Availability

The raw data is available as a Supplemental File.

## Supplemental Information

Supplemental information for this article can be found online at http://dx.doi.org/10.7717/peerj.6666#supplemental-information.

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
