# Peer review of "The relationship between serum uric acid within the normal range and β-cell function in Chinese patients with type 2 diabetes: differences by body mass index and gender"

_PeerJ, doi:10.7717/peerj.6666_

## Round 0.1 · original submission · Major Revisions

Authors should address the reviewers' concerns, including issues about reference ranges linked to race (BMI, HOMA indexes) and sex (SUA), and revaluate their results accordingly. Metabolic syndrome criteria should be better defined. Also, new concepts concerning the relationship between SUA levels and pharmacological therapies, such as Dapagliflozin, can be added to the discussion.

Reviewer 1 ·

Basic reporting

In this study the Authors aim to demonstrate the relationship between serum uric acid (SUA) within the normal range and the beta cell function (HOMA 2B) and insulin resistance indexes (HOMA IR) in type 2 diabetic Chinese population.

Experimental design

In my opinion, the Authors should also keep in mind that Asian population have different cut offs for BMI, HOMA-IR, HOMA 2B and other metabolic indexes. In this regard, it would be necessary to make a new additional table and to re-evaluate the association between SUA levels and cut off point of BMI recommended for the Asian population.

Validity of the findings

The results of this study evaluate for the first time the association between normal range of SUA with beta cell function and insulin resistance in diabetic Chinese population. It is interesting to note that normal SUA levels were associated with beta cell function in overweight/ obese people, while no association was seen in normal weight people. However, it would be attractive deepened even with further studies the molecular mechanisms underlying these observations of associations.

Reviewer 2 ·

Basic reporting

Not completely convincig results

Experimental design

Sample size adequate

Validity of the findings

I find a little bit contradictory that UA levels are associated with beta cells function and contextually do not mirror IR in T2DM patients.
The normal range of UA is different for gender.

The reference ranges for uric acid are as follows:
Men: 2.5-8 mg/dL
Women: 1.9–7.5 mg/d

Gender plays a key role also for other aspects as ... Cardiol. 2016 Feb;67(2):170-6.
Gender differences in the association between serum uric acid and prognosis in patients with acute coronary syndrome.

Additional comments

I find a little bit contradictory that UA levels are associated with beta cells function and contextually do not mirror IR in T2DM patients.
Probably, this is due to the imprecision of surrogate markers used for the assessment.......my hypothesis!
As the authors know very well.....The normal range of UA is different for gender and thus there are renal factors involved.
Authors should give values also in mg/dL for the sake of completeness.
The reference ranges for uric acid are as follows:
Men: 2.5-8 mg/dL
Women: 1.9–7.5 mg/dL
Gender plays a key role also for other aspects as ... Cardiol. 2016 Feb;67(2):170-6.
Gender differences in the association between serum uric acid and prognosis in patients with acute coronary syndrome.
The ACS is the main complication of T2DM patients,

·

Basic reporting

Good

Experimental design

Good

Validity of the findings

Good

Additional comments

The current study suggested that: 1) SUA levels are significantly associated with HOMA2%B in T2DM patients with overweight/obesity group, but not in normal weight group; 2) other islet function indexes, such as FCPI, PPCPI, and ΔC-peptide, did correlate with SUA levels in T2DM patients with overweight/obesity group; 3) the absence of a relationship between SUA and HOMA2IR after adjustment for Cr, BMI, sex, HbA1c, and diabetic duration in T2DM patients with overweight/obesity.
To define Metabolic Syndrome, the presence of at least three of the proposed criteria is required, and sometimes it is sufficient to have only one laboratory value, modified by diet or drugs, for the classification of Metabolic Syndrome. [1]. Recently, long term high fructose diet induced metabolic syndrome with increased blood pressure and proteinuria in rats [2]. Metabolic syndrome was associated with dual increase in renal glucose and uric acid transporters, including SGLT1, SGLT2, GLUT2, GLUT9 and UAT [2]. Dapagliflozin, a novel oral diabetic drug - Sodium-glucose cotransporter 2 inhibitor, could improve glycemic control and lower SUA levels in hospitalized patients with uncontrolled T2DM [3].
Authors are kindly requested to emphasize the current concepts about these issues in the context of recent knowledge and the available literature. This articles should be quoted in the References list.
References
1. What about non-alcoholic fatty liver disease as a new criterion to define metabolic syndrome? World J Gastroenterol. 2013 Jun 14; 19 (22): 3375-84. doi: 10.3748/wjg.v19.i22.3375.
2. Alterations of Renal Epithelial Glucose and Uric Acid Transporters in Fructose Induced Metabolic Syndrome. Kidney Blood Press Res. 2018;43(6):1822-1831. doi: 10.1159/000495814.
3. Effects of dapagliflozin on serum uric acid levels in hospitalized type 2 diabetic patients with inadequate glycemic control: a randomized controlled trial. Ther Clin Risk Manag. 2018 Dec 11;14:2407-2413. doi: 10.2147/TCRM.S186347.

---

## Round 0.2 · accepted · Accept

The authors have satisfactorily addressed the issues raised by the reviewers.

Reviewer 1 ·

Basic reporting

No further comments

Experimental design

No further comments

Validity of the findings

No further comments

Reviewer 2 ·

Basic reporting

Fine

Experimental design

Fine

Validity of the findings

Valid

Additional comments

Manuscript improved

·

Basic reporting

Good

Experimental design

Good

Validity of the findings

Good

Additional comments

Homeostatic model assessment for insulin - resistance (HOMA-IR), a relation between fasting glucose and insulin has been recognized as the most sensitive and specific method for measuring insulin resistance and risk for T2D, therefore Metabolic Syndrome. If it speaks of insulin - resistance the authors should specify that to define Metabolic Syndrome, the presence of at least three of the proposed criteria is required, and sometimes it is sufficient to have only one laboratory value, modified by diet or drugs, for the classification of Metabolic Syndrome [1].
Authors are kindly requested to emphasize the current concepts about these issues in the context of recent knowledge and the available literature. This article should be quoted in the References list.
Reference
1. What about non-alcoholic fatty liver disease as a new criterion to define metabolic syndrome? World J Gastroenterol. 2013 Jun 14; 19 (22): 3375-84. doi: 10.3748/wjg.v19.i22.3375.